# The Multifaceted Pyruvate Metabolism: Role of the Mitochondrial Pyruvate Carrier

**DOI:** 10.3390/biom10071068

**Published:** 2020-07-17

**Authors:** Joséphine Zangari, Francesco Petrelli, Benoît Maillot, Jean-Claude Martinou

**Affiliations:** Department of Cell Biology, Faculty of Sciences, University of Geneva, 30 Quai Ernest Ansermet, 1211 Geneva 4, Switzerland; Josephine.zangari@unige.ch (J.Z.); francesco.petrelli@unige.ch (F.P.); benoit.maillot@unige.ch (B.M.)

**Keywords:** mitochondria, mitochondrial pyruvate carrier, metabolism, neurodegeneration, metabolic disorders, cancer

## Abstract

Pyruvate, the end product of glycolysis, plays a major role in cell metabolism. Produced in the cytosol, it is oxidized in the mitochondria where it fuels the citric acid cycle and boosts oxidative phosphorylation. Its sole entry point into mitochondria is through the recently identified mitochondrial pyruvate carrier (MPC). In this review, we report the latest findings on the physiology of the MPC and we discuss how a dysfunctional MPC can lead to diverse pathologies, including neurodegenerative diseases, metabolic disorders, and cancer.

## 1. Introduction

Mitochondria are essential organelles of endosymbiotic origin, which participate in a multitude of cellular functions in eukaryotic cells, including energy metabolism, biosynthetic reactions, signaling, and the execution of programmed cell death. They host the respiratory chain complexes and ATP synthase, all of which participate in the formation of ATP (adenosine triphosphate) through the process known as oxidative phosphorylation (OXPHOS). Electrons released through oxidation of carbohydrates, amino acids, or lipids in the mitochondrial tricarboxylic acid (TCA) cycle are stored in ATP in the form of two energy-rich phosphoanhydride bonds. Of the molecules that can provide electrons to the respiratory chain, pyruvate, the end-product of glycolysis, is the most critical in many cell types including neurons. In mitochondria, oxidation of pyruvate by pyruvate dehydrogenase (PDH) generates acetyl coenzyme A (acetyl-CoA), which can then combine with oxaloacetate (OAA) to form citrate, the first substrate of the TCA cycle (Figure 1). 

Furthermore, pyruvate can be carboxylated by the pyruvate carboxylase (PC) to OAA (Figure 1), which represents a major anaplerotic pathway to replenish TCA cycle intermediates, not only for gluconeogenesis but also for other pathways including the urea cycle and lipid synthesis [1].

In addition to glycolysis, there are other, mostly minor sources of pyruvate such as oxidation of lactate by lactate dehydrogenase (LDH), conversion from alanine by alanine transaminase (ALT), or conversion from malate by cytosolic or mitochondrial malic enzyme (ME) (Figure 1). Once formed, pyruvate can be either reduced to lactate by LDH in the cytosol, regenerating NAD^+^ to fuel glycolysis, or fully oxidized within mitochondria, through the TCA cycle. The choice between these two pathways has important consequences for the cell, since glycolysis yields two molecules of ATP/molecule of glucose, whereas oxidative phosphorylation yields >30 molecules of ATP/molecule of glucose [2,3]. 

To enter mitochondria, pyruvate crosses the outer mitochondrial membrane (OMM) to reach the intermembrane space (IMS), probably through the large, relatively non-specific, voltage-dependent anion channel (VDAC), and it is then transported together with a proton across the inner mitochondrial membrane (IMM) by the mitochondrial pyruvate carrier (MPC) [4] (Figure 1). The existence of MPC was proposed on theoretical grounds several decades ago [4], although the molecular identification of the MPC complex was only achieved in 2012 [5,6]. As the sole point of entry for pyruvate into the mitochondrial matrix, the MPC plays a crucial role in coordinating glycolytic and mitochondrial activities, and it provides a key decision point for modulating cellular energy production and metabolism.

In this review, we report the most recent findings on the physiology of the MPC and its participation in various pathologies, including neurodegenerative diseases, metabolic disorders, and cancer.

## 2. Structure of the MPC

The MPC is encoded by three homologous genes *MPC1*, *MPC2*, and *MPC3* in *Saccharomyces cerevisiae*, by two genes *MPC1* and *MPC2* in flies, and by three genes, *MPC1*, *MPC1-like*, and *MPC2* in mammals [5,6]. In yeast, the active MPC complexes are the MPC1-MPC3 heterodimers, which promote pyruvate transport during respiratory growth, and the MPC1-MPC2 heterodimers, which function during fermentable growth [7,8]. In most mammalian cells, the active carrier is composed of an MPC1 and MPC2 heterodimer, with the exception of spermatocytes, which display MPC1-like and MPC2 heterodimers [9]. Loss of one subunit leads to degradation of the other subunit and disruption of the MPC complex. Functional tests with yeast MPC1 and MPC3 following reconstitution of the carrier in liposomes showed that only heterodimers were able to transport pyruvate [10]. In another report, MPC2 homodimers were also reported to be functional [11], although this was not supported by the results of Tavoulari et al. [10] or by other data reporting that mitochondria from ΔMPC1 mutants were unable to import pyruvate [5,6,12]. The reason for this discrepancy remains unclear.

MPC1 and MPC2 are small integral membrane proteins of, respectively, 12 kDa and 14 kDa. Structure predictions using different algorithms suggest that MPCs belong to the semi-SWEET (Sugar Will Eventually be Exported Transporter) domain family (SLC50 family) [13] or to the SWEET family [14], also known as the PQ-loop family of sugar transporters. The semi-SWEET domain is composed of a simple 1-3-2 triple transmembrane helix bundle (THB), whereas the SWEET domain consists of two semi-SWEET domains linked by a transmembrane helix [15] (Figure 2A). Recently, Medrano-Soto et al. proposed that the MPC belongs to the transporter-opsin-G protein-coupled receptor (TOG) superfamily with seven putative TMSs arranged in a 3+1+3 topology [14]. According to these authors, the MPC1 and MPC2 subunits might have originated from duplication of an MPC precursor, composed of four transmembrane segments, which would have lost its N-terminal transmembrane segment. Although the mechanisms via which the MPC imports pyruvate remain unknown, several hypotheses were postulated using in silico docking analyses, based on structural models of the MPC [16,17,18].

The heterodimeric composition and homology to the SWEET or semi-SWEET sugar transporters, sets the MPC apart from other families of mitochondrial carriers (named MCF or SLC25). The membrane topology of MPC1 and MPC2 is still not fully resolved and Figure 2 proposes a model based on the semi-SWEET motif. Our earlier biochemical approaches based on the accessibility to proteases or to thiol labeling suggest that MPC1 displays at least two transmembrane segments with the N- and C-termini projecting into the mitochondrial matrix, whereas MPC3 in yeast and MPC2 in mammals probably consist of three transmembrane helices, with the N-term in the matrix and the C-term in the intermembrane space [7]. It was recently reported that, despite this topology, yeast MPC2 and MPC3, both of which display an odd number of transmembrane segments with the N-term in the matrix, are nevertheless imported via the carrier import pathway which includes the receptor Tom70, TIM (Translocase of the Inner Membrane) chaperones, and the TIM22 complex [19,20], and not via the flexible presequence pathway as was previously predicted.

Several *MPC1* mutations resulting in disruption of the MPC complex or loss of transporter function were reported [21,22]. All of these mutations are accompanied by severe clinical symptoms and premature death.

Solving the three-dimensional structure of the MPC will be key to resolving the remaining uncertainties concerning the structure of the carrier, its membrane topology, and how it transports pyruvate across the IMM.

## 3. Regulation of MPC Expression

### 3.1. Transcriptional Regulation of MPC Expression in Yeast and in Mammalian Cells

As mentioned above, in *Saccharomyces cerevisiae*, MPC1 and MPC2 are expressed under fermentative conditions and form the MPC_FERM_ complex, while MPC1 and MPC3 are expressed under respiratory conditions and form the MPC_OX_ complex [7,23]. This switch is orchestrated at the level of transcription by the activity of the high osmolarity glycerol (HOG) mitogen-activated protein (MAP) kinase pathway. Accordingly, the *MPC3* gene promoter is bound directly by the Sko1 transcriptional repressor/activator, which is one of the core transcription factors mediating the transcriptional osmostress response downstream of the HOG1 MAP kinase [8].

In mammalian cells, although MPC1 and MPC2 subunits are ubiquitously expressed, their expression is particularly abundant in the heart, kidney, liver, brown adipose tissue, muscles, and brain [24,25,26]. A number of transcriptional regulatory mechanisms of the MPC genes were reported. Wang and colleagues showed that MPC1 is transcriptionally repressed in human prostate cancer cells by the chicken ovalbumin upstream promoter-transcription factor II (COUP-TFII), a member of the steroid receptor superfamily. MPC1 repression is part of the metabolic switch toward increased glycolysis, which promotes prostate cancer cell growth and invasion [27].

In contrast to COUP-TFII, peroxisome proliferator-activated receptor-gamma co-activator (PGC)-1 alpha (PGC-1α) was found to increase expression of MPC1 in human renal cell carcinoma [28] and cholangiocarcinoma [29]. Overexpression of PGC-1α strongly stimulated *MPC1* transcription through binding to the *MPC1* promoter, while depletion of PGC-1α by small interfering RNA (siRNA) suppressed MPC1 expression. The latter study showed that PGC-1α reversed the Warburg effect by upregulating expression of both MPC1 and pyruvate dehydrogenase E1 alpha 1 subunit (PDHA1), thus leading to enhanced mitochondrial metabolism. Transcriptional activation of MPC1 by PGC-1α was shown to be mediated by recruitment of the estrogen-related receptor alpha (ERRα), which bound the ERRα response element located in the proximal *MPC1* promoter region [28,30]. Inhibiting the activity of ERRα decreased expression of MPC1, interfered with pyruvate entry into mitochondria, and increased cellular reliance on glutamine oxidation and the pentose phosphate pathway (PPP) to maintain reduced NAD phosphate (NADPH) homeostasis [30].

More recently, it was shown that colon cancer cells can reprogram cell metabolism to coordinate proper cellular response to interferon-γ (IFNγ), a cytokine that plays a pivotal role in host antitumor immunity. Downregulation of MPC subunit expression via the signal transducer and activator of transcription 3 (STAT3) pathway attenuated IFNγ-mediated apoptosis of the colon cancer cells by preventing production of reactive oxygen species (ROS) [31]. Moreover, inhibition of STAT3-mediated transcription using the inhibitor Stattic partially reversed the inhibition of MPC1 and MPC2 expression and increased the antitumor efficacy of IFNγ.

In prostate adenocarcinoma, the androgen receptor (AR) is a hormone-responsive nuclear receptor transcription factor that coordinates anabolic processes to enable tumor proliferation through transcriptional regulation of metabolic pathways [32]. Massie and colleagues showed that the AR regulated MPC activity via direct transcriptional control of *MPC2*. In AR-driven prostate adenocarcinoma, MPC inhibition led to reduced OXPHOS, activation of the eukaryotic initiation factor 2 α (eIF2α)/activating transcription factor 4 (ATF4) integrated stress response, and increased glutaminolysis [33]. Importantly, in this experimental model, MPC inhibition by the small-molecule inhibitor MSDC-0160 suppressed tumor growth in vivo, suggesting that the MPC could be a potential therapeutic target for this type of cancer.

Finally, it was found that, in pancreatic cancer, MPC1 is transcriptionally suppressed by the histone lysine demethylase 5A (KDM5A) [34]. Elevated expression of KDM5A and downregulation of MPC1 correlated directly with pancreatic ductal adenocarcinoma progression.

Taken together, these data indicate that a number of different transcription factors can regulate the promoter activity of MPC1 and/or MPC2. How these factors regulate expression of the MPC genes, and how this regulation is integrated into the overall activity of cellular signaling pathways are interesting questions that remain to be further investigated.

### 3.2. Post-Translational Regulation of MPC Expression

Several studies described the post-translational regulation of MPC activity. Liang and colleagues found that MPC1 is acetylated on lysine residues K_45_ and K_46_, and that deacetylation of MPC1 by Sirt3 resulted in increased carrier activity [35].

MPC2 was also reported to be acetylated in a mouse type 1 diabetic heart model. However, in this case, acetylation appeared to stimulate activity of the transporter [36].

An indirect model of post-translational regulation was reported by the group of Kim et al. [37], who showed that, in hepatocarcinoma cells, the BH3 (Bcl-2 homology region 3)-only protein PUMA (p53 upregulated modulator of apoptosis), whose transcription depends on p53, can associate with MPC to disrupt dimer formation and MPC function. High expression levels of PUMA were correlated with decreased mitochondrial pyruvate uptake and increased glycolysis.

## 4. The MPC and Cell Metabolism

Cell metabolism can be defined as the ensemble of chemical reactions occurring in the cell, including anabolic reactions that convert nutrients into molecular building blocks, and catabolic reactions that convert nutrients into energy. By allowing the import of pyruvate into mitochondria, the MPC participates in both anabolic (synthesis of intermediary metabolites by the TCA cycle) and catabolic (OXPHOS) events.

One of the characteristics of cell metabolism is its plasticity, which results from the interconnectivity between different metabolic pathways. Plasticity ensures that, when one pathway is transiently or permanently interrupted due to lack of an essential metabolite or a key enzyme or transporter, the cell is able to switch to an alternative pathway to compensate for the defect. However, bottlenecks exist, involving certain metabolite carriers, which allows metabolites to cross cellular membranes. Pyruvate metabolism is a good example. As mentioned in the introduction, there are several ways to synthesize this metabolite in the cytosol; however, the MPC is the only route via which pyruvate can cross the IMM. Such molecules provide key regulatory steps for the cell, as well as offer attractive targets for therapeutic intervention.

Nevertheless, because of the metabolic plasticity, cells often find ways to compensate for defective carriers or metabolites. For example, when the MPC is deficient, increased glutaminolysis or oxidation of fatty acids (beta-oxidation) can compensate for the deficit in pyruvate-derived carbon to fuel the TCA cycle [25,38,39].

Different cell types adapt their metabolism differently to changes in their cellular environment and physiology. In cells with high energy demands such as muscle cells and neurons, OXPHOS is the preferred source of ATP. For these cells, either glucose/pyruvate or fatty acids (FAs) provide the main substrates, although neurons preferentially oxidize pyruvate because they do not express many of the enzymes required for beta-oxidation. In contrast, cardiomyocytes in the adult seem to favor FA oxidation over pyruvate [40,41]. Thus, in these cell types, metabolic plasticity is more limited, which may explain the high incidence of degenerative pathologies that affect brain and muscle.

In other cell types, ATP is generated mainly through glycolysis rather than OXPHOS, even when oxygen is available. This is the case for many highly proliferating cells, including antigen-activated lymphocytes [42] and most cancer cells in vitro. This type of metabolism is referred to as aerobic glycolysis or the Warburg effect after Otto Warburg who first described this process in cancer cells [43,44].

In highly proliferating cells, an important advantage of glucose fermentation compared to OXPHOS is that carbohydrates are only partially degraded, thus providing intermediary metabolites, particularly nucleotides generated via the pentose pathway [43], as building blocks to maintain rapid growth.

Thus, it is clear from the above that the importance of the MPC in cell metabolism is highly dependent on the cell type and cell context. Below, we review the role of the MPC in different cell types and discuss how a dysfunctional MPC can lead to diverse pathologies.

### 4.1. The Role of MPC in Neurons and in Neurogenerative Diseases (NDs)

Neurons rely mainly on glucose as an energy source and to a lesser extent on amino-acid oxidation; however, they lack most enzymes involved in FA oxidation [45]. As a result, the role of the MPC, and of pyruvate metabolism in general, is particularly important in neurons. Pyruvate is generated principally through glycolysis in these cells, but also by conversion from lactate by LDH. Astrocytes provide the major source of lactate which is taken up by neurons by the so-called astrocyte-neuron shuttle [46].

Although most neurons use oxidative phosphorylation to generate the high levels of energy required for neural transmission, some parts of the brain, such as the medial and lateral parietal and prefrontal cortices, were found to rely mainly on aerobic glycolysis [47]. Energy metabolism in the retina is also predominantly through aerobic glycolysis, and only a small fraction of the pyruvate produced by glycolysis is oxidized in mitochondria. Nevertheless, pyruvate oxidation in mitochondria appears to be essential for retinal function since mice lacking MPC1 in the retina were found to display degeneration of both rod and cone photoreceptors and decline in visual function [48]. MPC-deficient retinas displayed lower ATP and NADH levels although increased glutaminolysis and ketone body oxidation limited degeneration of the photoreceptors.

Neurodegenerative diseases (NDs) are often considered to be metabolic disorders characterized by a decline in the ability to import or metabolize energy sources, resulting either in or from mitochondrial dysfunction [49]. However, even though bioenergetic defects were observed in diverse pathological conditions, both in mice and in patients [50,51,52], in most cases, it remains unclear whether this is the cause or the consequence of the pathology. Interestingly, it was shown recently that modulation of energy metabolism through MPC inhibition offers a potential pharmacological approach to treatment for NDs, in particular for Parkinson’s and Alzheimer’s diseases.

Parkinson’s disease (PD) is a neurodegenerative disorder resulting in the death of dopaminergic neurons in the substantia nigra pars compacta of the brain. In recent years, epidemiological evidence showed similarities in metabolic dysfunction between type 2 diabetes and PD. Indeed, several clinical trials in PD patients are in progress using anti-diabetic drugs [53,54,55,56,57,58,59]. In 2016, Ghosh and colleagues (2016) investigated the activity of the MPC inhibitor MSDC-0160, a derivative of thiazolidinedione, in diverse models of PD [60]. The authors showed that MSDC-0160 protected tyrosine hydroxylase (TH)-positive dopaminergic neurons against the neurotoxicity of MPP^+^ or MPTP in vitro and in vivo, as well as showed beneficial effects in the Engrailed heterozygous mutant mice, which undergo loss of dopaminergic neurons at six weeks of age [60,61]. The mechanisms via which MPC inhibition results in neuronal protection are not well understood. Ghosh and colleagues proposed that inhibition of the MPC may protect neurons through modulation of the mammalian target of rapamycin (mTOR) pathway and autophagy, and indeed MSDS-0160 was shown to reduce neuroinflammation by modulating the NF-κB (nuclear factor kappa-light-chain-enhancer of activated B cells)/mTOR pathway [60]. This could be especially relevant for PD since neuroinflammation is considered to play an important role in the pathophysiology of the disease [62,63].

MPC inhibition could also be beneficial in the treatment of Alzheimer’s disease (AD). A phase IIa clinical study in non-diabetic subjects with mild to moderate AD demonstrated that a three-month treatment with the MPC inhibitor MSDC-0160 resulted in a significant increase of glucose uptake in the regions of the brain normally affected in this pathology [64]. This again argues in favor of a possible neuroprotective effect of this compound. However, these results should be considered in the light of recent findings in vitro in which the expression of MPC2 was shown to be decreased in AD-related models [65]. This led to decreased calcium and pyruvate uptake into mitochondria and decreased OXPHOS. The reason for the decrease in calcium import remains unclear. However, another study in hepatocytes and embryonic fibroblasts showed that calcium import into mitochondria through the mitochondrial calcium uniporter (MCU) was decreased following inhibition of MPC activity. This effect was mediated by increased expression of the MCU gatekeeper mitochondrial calcium uptake 1 (MICU1) [66]. It would be interesting to test the expression levels of MICU1 in MPC-deficient neurons.

Another study describing the effects of pharmacological inhibition of the MPC reported that two specific inhibitors, MSDS-0160 and UK5099, protected neurons against glutamate-induced excitotoxicity in vitro [67]. This suggests that inhibition of the MPC could be useful in acute pathologies of the brain such as stroke or brain trauma where neurons are frequently exposed to toxic levels of glutamate.

Despite these promising results in PD, AD, and acute neuronal death, the molecular mechanisms underlying these effects remain to be elucidated. Furthermore, most studies showing a beneficial effect of MPC inhibition in these pathologies were based on small-molecule derivatives of thiazolidinediones. Although these compounds were shown to act as MPC inhibitors, off-target effects cannot be excluded. Therefore, it will be important to confirm these results using other chemical classes of MPC inhibitors or using genetic approaches in mouse models in which MPC1 or MPC2 are deleted specifically in neurons and/or glial cells.

### 4.2. The Role of MPC in Metabolic Disorders

Many studies reported that disruption of the MPC affects gluconeogenesis, a process known to play a role in the pathogenesis of type 2 diabetes (T2D) [68]. T2D can result from the dysfunction of several organs, including pancreas, liver, muscle, and kidney, all of which express the MPC. It will, therefore, be important to analyze the consequences of MPC downregulation in each of these organs.

#### 4.2.1. MPC in Pancreas

Glucose is an important physiological stimulus for insulin secretion by pancreatic β-cells. Elevated blood glucose triggers increased glucose uptake into these cells, synthesis of ATP by OXPHOS, and closure of the plasma membrane ATP-sensitive potassium channels (K_ATP_ channel), which then leads to membrane depolarization, entry of calcium, and insulin secretion. This phenomenon, termed glucose-stimulated insulin secretion (GSIS), requires both pyruvate oxidation and carboxylation in pancreatic β-cells [69,70,71]. Inhibition of the MPC, either pharmacologically using UK5099 or genetically using siRNAs directed against MPC1 or MPC2, reduced GSIS in the 832/13 cell line derived from INS-1 rat insulinoma cells, as well as in rat and human islets [72]. Oxygen consumption, the ATP/ADP ratio, and the NADPH/NADP^+^ ratio were all reduced upon inhibition of the MPC. Similar results were obtained in mice displaying inactive, truncated MPC2 [26], in mice carrying a targeted deletion of MPC2 in pancreatic cells, as well as in *Drosophila* [73]. All these experiments show that the MPC plays an important and evolutionarily conserved role in insulin-secreting cells through mediating glucose sensing, regulation of insulin secretion, and control of systemic glycemia.

#### 4.2.2. MPC in Liver

##### Gluconeogenesis

One of the mechanisms via which the liver participates in T2D is through gluconeogenesis [68,74], a major regulatory process in which non-carbohydrate substrates are converted into either free glucose or glycogen (Figure 1). Gluconeogenesis can also take place in the kidney, albeit to a lesser extent.

Hepatic glucose production is a critical physiological process that is required for maintaining normoglycemia during periods of nutrient deprivation. The major non-carbohydrate precursors are lactate, amino acids, and glycerol. Lactate is of particular importance. It is converted into pyruvate by the action of lactate dehydrogenase LDHB, and pyruvate is then imported through the MPC into mitochondria where it is carboxylated into OAA and reduced into malate. Malate is then exported into the cytosol where it can be used to generate glucose through several steps, including the reversible steps of glycolysis (Figure 1).

Lactate for gluconeogenesis is derived mainly from muscle cells undergoing anaerobic glycolysis. Under these conditions, muscle cells release lactate, which is then taken up by the liver to provide a substrate for gluconeogenesis. The glucose produced by the liver can in turn provide an energy source for muscle cells, thereby completing the cycle. This pathway, known as the Cori cycle or the glucose-lactate cycle, can account for up to 40% of plasma glucose turnover.

As expected, liver-specific deletion of MPC1 or MPC2 in mice impairs lactate/pyruvate-triggered hepatic gluconeogenesis [75,76]. However, gluconeogenesis from alanine was increased in MPC-deficient mice, and McCommis et al. [76] suggested that intramitochondrial transamination of alanine to pyruvate may contribute to gluconeogenesis when mitochondrial pyruvate import is inhibited. Thus, pyruvate-alanine cycling may constitute an alternative pathway for gluconeogenesis, which circumvents the MPC. This interesting hypothesis implies the existence of a mitochondrial transporter for alanine, which remains to be identified.

By decreasing gluconeogenesis, MPC inhibition was found to attenuate the development of hyperglycemia induced by a high-fat diet (HFD) leading to improved glucose tolerance [75,77].

##### MPC in Nonalcoholic Steatohepatitis (NASH)

The rise in the level of obesity in the population dramatically increased the incidence of a variety of related metabolic diseases, including nonalcoholic fatty liver disease (NAFLD). The spectrum of NAFLD ranges from simple hepatic fat accumulation to a more severe disease termed nonalcoholic steatohepatitis (NASH), involving inflammation, hepatocyte death, and fibrosis.

As mentioned above, thiazolidinediones (TZDs) appear to be potent MPC inhibitors [78]. One TZD derivative, MSDC-0602, prevents and reverses stellate cell activation and fibrosis in a mouse model of NASH. Importantly, the effects of this small-molecule inhibitor were duplicated by genetic deletion of MPC2 in hepatocytes and furthermore, the effects of MSDC-0602 were lost when MPC2 was deleted [79]. Thus, the MPC appears to be an attractive target in NASH [80]. Indeed, results from a phase IIb clinical trial on patients with liver biopsy-confirmed NASH [81] showed that MSDC-0602K significantly decreased liver steatosis, although it failed to prevent liver fibrosis.

#### 4.2.3. MPC in Kidney

Diabetic kidney disease (DKD), which is characterized by albuminuria and renal hypertrophy, is the leading cause of kidney failure. It was shown recently that treatment with artemether, a methyl ether derivative of artemisinin used in the treatment of malaria and identified as a possible candidate for treating T2D, prevented kidney hypertrophy and ameliorated the lesions that lead to renal enlargement in T2D db/db mice. Interestingly, the mechanisms underlying this beneficial effect may in part be associated with the ability of artemether to increase MPC1 and MPC2 levels in db/db mice [82]. In particular, podocytes, which play an important role in the development of DKD, undergo increased apoptosis when the MPC is inhibited using UK5099 or RNA interference. These results suggest that enhancing MPC function may reduce injury in high-glucose-treated podocytes and may possibly attenuate DKD.

#### 4.2.4. MPC in Muscle

During T2D, decreased glucose uptake by skeletal muscle significantly drives chronic hyperglycemia [83]. Skeletal muscle-specific MPC knockout in mice (MPC SkmKO) leads to increased glucose uptake in muscle and increased lactate release [84], thus increasing the Cori cycle as explained above. Furthermore, FA oxidation appears to be increased in MPC-deficient cells. Because hepatic gluconeogenesis is energetically supported by FA oxidation and muscle MPC disruption increases muscle FA oxidation, futile Cori cycling is energetically supported by FA oxidation. In conclusion, these findings raise the possibility that selectively decreasing skeletal muscle pyruvate uptake in obese and T2D patients may promote fat loss and restoration of whole-body insulin sensitivity.

In conclusion of this part on metabolism and pathologies, it emerges that, while inhibition of the MPC in complex metabolic diseases, such as T2D, is potentially interesting, it is difficult to predict the outcome in patients for two main reasons: (i) because these pathologies are the result of multiorgan dysfunction, and (ii) because antagonistic effects of MPC inhibition occur in different organs. For example, while MPC inhibition increases glucose uptake in the muscles and decreases gluconeogenesis in the liver, two beneficial effects for T2D, it also decreases insulin secretion, which a priori could be problematic for diabetic patients. Thus, it remains unclear whether the combination of these positive and negative effects will result in an overall benefit for the patient.

Due to embryonic lethality in mice at around E12, it is not possible to generate a constitutive deletion of either MPC1 or MPC2 in all tissues [24,25,26] and, to date, experiments addressing the role of MPC in the mouse were mainly performed using organ-specific deletion of the carrier. Furthermore, we do not yet know whether a global, conditional knockout of MPC1 or MPC2 in adult mice would be lethal. An alternative approach, therefore, is to address this question pharmacologically by administering small-molecule inhibitors of the MPC, with the caveat that off-target effects cannot be excluded. Such experiments were performed recently and, as described above, promising results were obtained in mouse models of NASH and T2D.

### 4.3. The Role of MPC in Cancer

As described above, cancer cells in vitro rely mainly on aerobic glycolysis for rapid cell growth (Figure 3), and it is not surprising, therefore, that loss of MPC expression, which favors increased glycolysis, was found to be associated with poor cancer prognosis [85,86,87,88,89,90].

One of the main questions in the field is whether loss of functional MPC can initiate tumorigenesis and, conversely, whether increasing MPC activity would result in reduced aerobic glycolysis. Furthermore, if aerobic glycolysis is generally associated with tumor growth, there are a number of reports showing that OXPHOS and ROS production are required for tumor metastasis. In this case, what is the role of the MPC? Does expression of the MPC contribute to poor prognosis, and would inhibition of the MPC be able to reduce cell invasion and metastasis? These are the questions we will address in the next section.

#### 4.3.1. The Role of MPC in Stemness

Re-expression of wild-type (WT) MPC1 and MPC2 in colon cancer cells, which carried mutations or deletions in the MPC1 gene, impaired colony formation in soft agar and spheroid formation in vitro and reduced tumor growth in vivo [89]. In addition, these antitumoral effects were accompanied by a decrease in stem-cell markers, such as aldehyde dehydrogenase (ALDH) A (ALDHA), lin-28 homolog A (LIN28A), leucine rich repeat containing G protein-coupled receptor 5 (LGR5), and homeobox transcription factor Nanog (NANOG), suggesting that a decrease in MPC expression promotes the Warburg effect and the maintenance of stemness in colon cancer cells. Consistent with these findings, Zhenhe Suo and colleagues showed that pharmacological or genetic inhibition of the MPC stimulated aerobic glycolysis in prostate, esophageal squamous, and ovarian cancer cells in vitro [87,91,92,93]. This was associated with higher levels of stem-cell markers including organic cation transporter (OCT) OCT3/4, NANOG, hypoxia inducible factor 1 alpha (HIF1α), notch receptor 1 (NOTCH1), CD44 antigen, and ALDH [91,93]. Moreover, MPC inhibition conferred on these cells the ability to migrate and an increased resistance to both chemotherapy and radiotherapy [87,91,93]. Similar observations were reported for lung adenocarcinomas [90]. Taken together, these results highlight a role of MPC in determining the stemness status of cancer cells in vitro.

More recently, MPC was shown to be involved in the initiation of intestinal tumor formation in mice and *Drosophila* [94]. Sporadic colon tumors are believed to follow a typical progression pathway in which the initial tumorigenic event triggers intestinal stem-cell hyperplasia, leading to the formation of a benign adenoma. This initial event was associated with hyperactivation of the Wnt/β-catenin pathway and loss of function of the *APC* tumor suppressor. In this study, expression of both MPC1 and MPC2 was found to be decreased in adenomas in two different mouse models of colon cancer, the azoxymethane and dextran sodium sulfate (AOM-DSS) models, as well as a genetic model of heterozygous loss of *Apc* in intestinal stem cells (*Apc^Lrig1KO/+^Mpc1^Lrig1KO^* mice) [94]. Targeted deletion of *MPC1* in adult LRIG1^+^ intestinal stem cells (*Mpc1^Lrig1KO^*) led to an increased tumor burden and a substantial increase in the incidence of macroscopic tumors in the AOM-DSS model. Furthermore, in the heterozygous *Apc* model, the *Apc^Lrig1KO/+^Mpc1^Lrig1KO^* mice exhibited a higher tumor burden compared to *Apc^Lrig1KO/+^* mice, and the macroscopic tumor burden was also much higher. These results demonstrate that loss of MPC function is sufficient to promote intestinal tumor initiation in a chemically induced tumor model and in different genetic tumor models. Similarly, in *Drosophila*, loss of the MPC or *Apc* led to hyperproliferation of intestinal stem cells and, importantly, the hyperproliferation following deletion of *Apc* was completely suppressed by ectopic expression of the MPC. Interestingly, the metabolic consequences of MPC loss resulted in all *Mpc1^Lrig1KO^* adenomas attaining a stem-like phenotype.

In conclusion, decreased mitochondrial pyruvate metabolism through elimination of MPC activity is sufficient to increase the oncogenic susceptibility of both the fly and the mouse intestinal tracts. Thus, constitutive enforcement of the metabolic program found in hyperproliferative colonic lesions predisposes intestinal stem cells to adenoma formation.

#### 4.3.2. The Role of MPC in Epithelial-Mesenchymal Transition (EMT)

During tumorigenesis, cancer cells acquire migratory and invasive properties through induction of the epithelial-mesenchymal transition (EMT). It was proposed that production of ROS can trigger metastasis [95], and some authors proposed that aerobic glycolysis would promote primary tumor formation, while a shift to a more oxidative metabolism would be required for metastasis (Figure 3). In intrahepatic cholangiocarcinoma, known to have a high malignant potential, low *MPC1* expression is correlated with poor prognosis and a significant increase in the percentage of distant metastasis [88]. The most well-known phenomenon associated with metastasis of cancer cells is EMT, which is strongly linked to the function of MPC1. Indeed, MPC1 expression was downregulated in human biliary tract cancer cells undergoing TGF-β (transforming growth factor beta)-induced EMT, and the knockdown of MPC1 expression led to induction of EMT in these cancer cells. These findings support the conclusion that MPC1 functions as a modulator of EMT induction and contributes to the malignant potential of intrahepatic cholangiocarcinoma cells.

Consistent with this conclusion, the study of Takaoka and collaborators found that, in pancreatic and colon cancer cells, EMT was induced following suppression of MPC1 expression [96]. These authors showed that MPC1 and MPC2 knockdown upregulated the glutaminase GLS, inducing EMT. This effect was suppressed in glutamine depletion conditions, revealing glutamine metabolism as an important mechanism inducing EMT. Finally, *MPC1* was found to be downregulated in renal cell carcinoma tissue when compared with adjacent non-cancerous tissue, and lower MPC1 expression correlated with unfavorable prognosis for renal cell carcinoma patients [97]. Functionally, MPC1 suppressed the invasion of renal carcinoma cells in vitro and reduced their growth in vivo by decreasing the expression of the matrix metalloproteases 7 and 9 (MMP7 and MMP9).

#### 4.3.3. MPC and Lactate in Tumor Growth

The consequence for cancer cells of relying on aerobic glycolysis is, firstly, the need to import high levels of glucose to compensate for the loss of ATP production by OXPHOS and, secondly, high amounts of lactate produced in the cytosol by LDH-mediated reduction of pyruvate are released from the cell. The latter leads to acidification of the extracellular microenvironment, which favors metastasis, angiogenesis, and immunosuppression [98]. Thus, lactate can be seen as an oncometabolite in the metabolic reprogramming of cancer cells. However, the role of lactate in cancer is complex. Recently, two groups reported high levels of lactate in the blood of patients with lung cancers [99,100], as well as in a mouse lung cancer model [100]. Interestingly lactate was found to be imported into tumor cells to at least the same extent as glucose, although the consequences of high levels of lactate in cancer cells remains unclear. In particular, it is unclear whether lactate, after conversion into pyruvate, could fuel the TCA cycle, participate in OXPHOS, and/or lead to the synthesis of intermediary metabolites, including acetyl-CoA, which could influence chromatin remodeling and gene expression.

#### 4.3.4. Inhibition of MPC Activity Delays Tumor Growth

Reduced levels and activity of the MPC are associated with the majority of cancer types. Therefore, in most cases, a therapeutic strategy based on MPC would need to promote MPC expression and/or its activity. However, there are two reports to date of tumors in which inhibition of the MPC was shown to delay tumor growth, as well as a third report showing a promising adjuvant effect of MPC inhibition when coupled with radiotherapy.

In one report (see Section 3.1) the androgen-sensitive prostate tumor was shown to require MPC for growth and it should, therefore, be sensitive to MPC inhibition [33]. In a second report, it was shown that liver-specific disruption of MPC in mice decreased development of a hepatocellular carcinoma induced by a low-dose exposure to *N*-nitrosodiethylamine (DEN) plus carbon tetrachloride (CCl4) [101]. In the latter case, MPC-disrupted hepatocytes showed increased glutaminolysis to maintain the TCA cycle, and re-synthesis of glutathione was found to occur at a lower rate because less glutamine was available for glutathione synthesis. These findings raise the possibility of a model in which inducing metabolic competition for glutamine by MPC disruption would impair hepatocellular tumorigenesis by limiting glutathione synthesis. In the third report, MPC inhibition led to decreased oxygen consumption by tumor cells, thereby sparing oxygen locally and reducing hypoxia in the vicinity of the tumor. Importantly, this higher oxygen concentration around cells exacerbated the toxic effects of radiotherapy [102].

All together these results suggest that MPC inhibitors could be useful therapeutically to treat some selected cancer types.

## 5. Conclusions

In conclusion, mitochondrial pyruvate transport is essential for normal embryonic development and plays a key role in the function of many organs in the adult. Being at the heart of cell metabolism, MPC activity is solicited in several processes that require the presence of pyruvate inside mitochondria, to drive either cataplerotic reactions, such as gluconeogenesis or lipid synthesis, or anaplerotic reactions, to drive TCA cycle activity and consequently OXPHOS. Although cells can adapt their metabolism to circumvent impaired metabolic changes, this does not always prevent perturbation of cell homeostasis and pathology. We can see that, if preserving or restoring MPC activity is the objective in several pathologies, including certain cancers, we can also show that inhibition of MPC activity could be beneficial in some pathologies. These pathologies, which include T2D, NASH, and PD, as well as probably others yet to be discovered, may provide diverse therapeutic applications for MPC inhibitors.

## Figures and Tables

**Figure 1 biomolecules-10-01068-f001:**
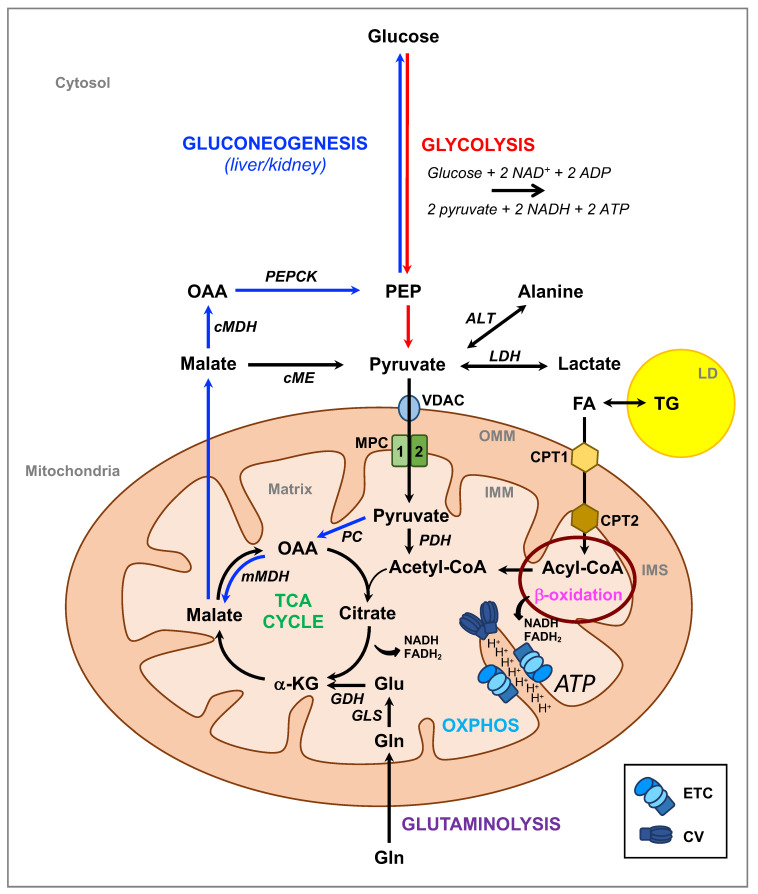
Metabolic pathways involving mitochondria. In the cytosol, pyruvate is produced through glycolysis, which generates two adenosine triphosphate (ATP) and two reduced nicotinamide adenine dinucleotide (NADH) molecules per molecule of glucose. Pyruvate can also be produced from oxidation of lactate by lactate dehydrogenase (LDH), conversion from alanine by alanine transaminase (ALT), or from malate by cytosolic or mitochondrial malic enzyme (ME). Pyruvate can be imported into mitochondria to be oxidized into acetyl coenzyme A (acetyl-CoA) by the pyruvate dehydrogenase (PDH), which then fuels the tricarboxylic acid (TCA) cycle. Import of pyruvate requires the voltage-dependent anion channel (VDAC) to cross the outer mitochondrial membrane (OMM) and the mitochondrial pyruvate carrier (MPC) to cross the inner mitochondrial membrane (IMM). The TCA cycle can also be fueled by glutamine (Gln) through glutaminolysis or by fatty acids (FA) released from lipid droplets (LD) where they are stored in the form of triglycerides (TG). FAs provide acetyl-CoA through FA β-oxidation. The TCA cycle and β-oxidation both generate the reducing equivalents NADH and flavin adenine dinucleotide (FADH_2_), which transfer electrons to the electron respiratory chain (ETC), generating more than 30 ATP molecules per molecule of glucose. The last reaction is catalyzed by ATP synthase or Complex V (CV). This process requires the presence of oxygen and is known as oxidative phosphorylation (OXPHOS). In the liver and kidney, pyruvate can be converted into oxaloacetate (OAA) by the pyruvate carboxylase (PC), which is then reduced into malate by the malate dehydrogenase (mMDH). Malate is then exported into the cytosol, converted into OAA by malate dehydrogenase (cMDH) and into phosphoenolpyruvate (PEP) by PEP carboxykinase (PEPCK) and from there into glucose through several steps, including the reversible steps of glycolysis. IMS: intermembrane space; CPT1: carnitine palmitoyltransferase 1; CPT2: carnitine palmitoyltransferase 2; GDH: glutamate dehydrogenase; GLS: glutamase; α-KG: α-ketoglutarate.

**Figure 2 biomolecules-10-01068-f002:**
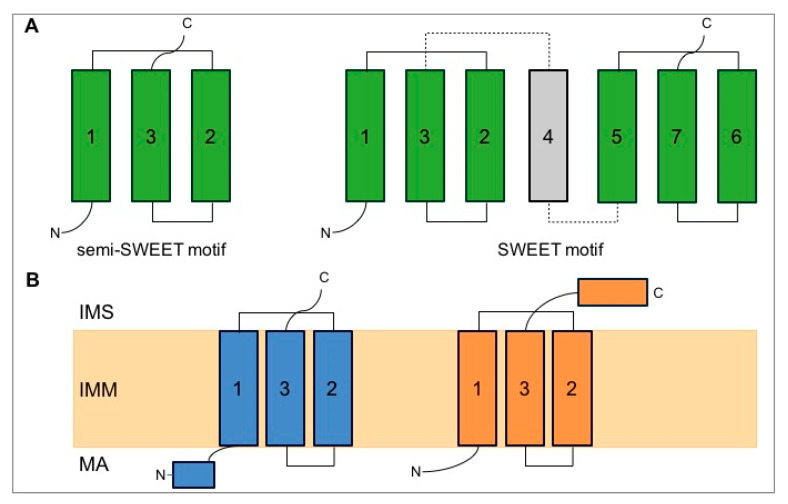
(**A**) Semi-SWEET and SWEET motifs. Semi-SWEET is composed of a triple helix bundle in this specific order 1-3-2. SWEET contains two semi-SWEET motifs linked together by a helix. All the helices are crossing the membrane. (**B**) Model for the topology of MPC1 and MPC2 in the inner mitochondrial membrane (IMM). Here, a semi-SWEET structure is proposed for both MPC1 and MPC2. MA: matrix; IMS: intermembrane space.

**Figure 3 biomolecules-10-01068-f003:**
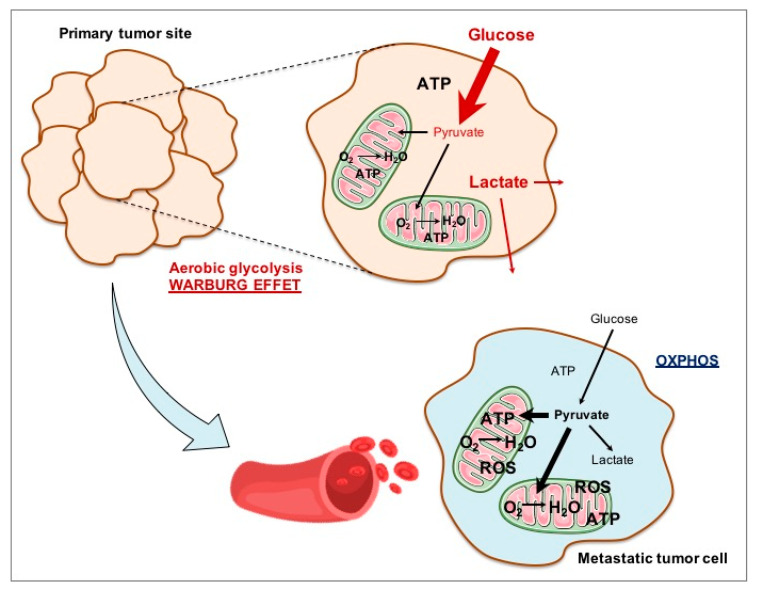
Tumor cell metabolism. In the primary tumor site, cancer cells mainly rely on aerobic glycolysis (Warburg effect) and this metabolism favors cell proliferation and tumor growth. Some cancer cells escape from the primary tumor site through the bloodstream, generating metastasis in other organs. It is thought that, to do this, some cancer cell types switch to a more oxidative metabolism.

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
