# Peer review of "The Multifaceted Pyruvate Metabolism: Role of the Mitochondrial Pyruvate Carrier"

_biomolecules, 2020, doi:10.3390/biom10071068_

Round 1

Reviewer 1 Report

This is a concise review and update on the status of the mitochondrial pyruvate carrier.  It is timely for updating work showing clinical relevance and promise for novel therapies.  I believe the authors could balance reference to only their work on the modeling the structure of MPC.  Here are four articles that should be read and studied to properly update the status on the hypothetical structure and binding activities of MPC.  The authors self-citation and this 2020 report use the SWEET or SemiSWEET as a template for homology modeling, but it is not a proton transporter.  Regardless, the findings are quite similar; however, much greater detail on the role of cysteine residues within the transporter channel are considered in greater detail in the earlier papers.  These detailed analyses of the homology model using the membrane domain of respiratory complex I from E. coli as the template for the human MPC monomers, are used to compare back to the earliest reports on the MPC by the classic work of Nałȩcz and Halestrap.

These works should be worthy of a brief mention at least.

Lee, J.; Jin, Z.; Lee, D.; Yun, J.-H.; Lee, W. Characteristic Analysis of Homo- and Heterodimeric Complexes of Human Mitochondrial Pyruvate Carrier Related to Metabolic Diseases. Int. J. Mol. Sci. 2020, 21, 3403.

Phelix, C. F.; Bourdon, A. K.; Dugan, J. L.; Villareal, G.; Perry, G. MSDC-0160 and MSDC-0602 Binding with Human Mitochondrial Pyruvate Carrier (MPC) 1 and 2 Heterodimer: PPARγ Activating and Sparing TZDs as Therapeutics. International Journal of Knowledge Discovery in Bioinformatics (IJKDB) 7(2)

Bourdon, A. K.; Villareal, G.; Perry, G.; Phelix, C. F.  Alzheimer's and Parkinson's Disease Novel Therapeutic Target: The Mitochondrial Pyruvate Carrier - Ligand Docking to Screen Natural Compounds Related to Classic Inhibitors  International Journal of Knowledge Discovery in Bioinformatics (IJKDB) 7(2)

Dugan, J. L.; Bourdon, A. K.; Phelix, C. F. Mitochondrial Pyruvate Carrier 1 and 2 Heterodimer, In Silico, Models of Plant and Human Complexes: A Comparison of Structure and Transporter Binding Properties. International Journal of Knowledge Discovery in Bioinformatics (IJKDB) 7(2)

Author Response

We thank the Reviewer for her/his time and efforts to consider our manuscript and to offer insightful comments on our review. We re-worded the text accordingly. All changes in the revised manuscript are in red.

We fully agree with the Reviewer that the structure and binding activities of the MPC are largely hypothetical. As suggested by the Reviewer we have now quoted the four articles  he mentioned in her/his comment (see end of page 4). In doing so, we bring all the models discussed in these papers to the attention of all readers interested in the structure of the MPC.

Reviewer 2 Report

The review "The multifaceted pyruvate metabolism; role of the mitochondrial pyruvate carrier" (manuscript biomolecules-855778) by Zangari et al. discuss the latest findings on the physiological role of the mitochondrial pyruvate carrier. The review is well written, and the subject is thoroughly analyzed. The manuscript is recommended for publication in Biomolecules given that the below minor comments are dealt with.

Figure legend of Fig. 1. It should be in place to spell out all the abbreviations used in the figure (including IMS, CPT1, CPT2 GDH and GS).

Line 55-58 mentions the number of ATP produced from glucose in glycolysis and oxidative phosphorylation. This statement should be referenced.

Fig. 2 with the topology of the SWEET and semi-SWEET motifs. Maybe the predicted sides of the membrane should be indicated; out/in, IMS/matrix or just cytoplasm.

The authors are advised to initially discuss the relative expression levels of MPC1 and MPC2 in different tissues before starting to talk about the regulation of their expression.

Author Response

We thank the Reviewer for her/his time and efforts to consider our manuscript and to offer insightful comments on our review. We re-worded the text accordingly.

All changes in the revised manuscript are in red.

Comments and answers:

Figure legend of Fig. 1. It should be in place to spell out all the abbreviations used in the figure (including IMS, CPT1, CPT2 GDH and GS).

All the abbreviations have been spelled out. They are in red in the text (pages 2 and 3).

Line 55-58 mentions the number of ATP produced from glucose in glycolysis and oxidative phosphorylation. This statement should be referenced.

The statement has been referenced (page 3)

Fig. 2 with the topology of the SWEET and semi-SWEET motifs. Maybe the predicted sides of the membrane should be indicated; out/in, IMS/matrix or just cytoplasm.

Figure 2 has been modified with a model for the topology of MPC in Figure 2B. IMM,IMS and OM are mentioned on this part of the figure. See page 5.

The authors are advised to initially discuss the relative expression levels of MPC1 and MPC2 in different tissues before starting to talk about the regulation of their expression.

A sentence mentioning that the MPC is mainly expressed in the heart, kidney, liver brown fat tissue, muscles and brain has been added with 3 references in the revised manuscript (page 6)

Reviewer 3 Report

In the review by Zangari and colleagues, the mitochondrial pyruvate carrier (MPC) complex is discussed with particular focus given to the structure of the MPC, expression, and role in metabolism and disease. Interest in this carrier has exploded since the authors discovered it in 2012. This review provides extensive coverage of the literature on this topic. There are a few requested additions to improve the paper.

-In the first paragraph of the introduction, there is discussion of pyruvate oxidation, but no mention of carboxylation. Anaplerosis is also an important mitochondrial fate of pyruvate.

-Line 77 -  the use of the word “variant” to describe MPC1-like is probably not the best choice since it is encoded by a separate gene. Also, this seems to be at odds with the discussion just above it indicating there are only two MPC genes in mammals.

-There should be some mention that deletion of one isoform of MPC results in the degradation of the other, i.e. knockout of one isoform essentially results in a double knockout.

-The SWEET and semiSWEET schematic should be replaced with a more informative MPC topographical schematic.

-In the discussion of the topology, some of the more recent literature should be discussed such as Gomkale et al 2020; and Medrano-Soto 2020. It seems as though there is a growing consensus.

-The statement that “MPC in liver is required for gluconeogenesis” (line 322) seems to be too strong.  While the MPC is important for gluconeogenesis, both of the references (Ref 66, 67) show that mice are still able to maintain a degree of gluconeogenic capacity from other mechanisms.

-Recent results from the EMMINENCE trial should be included since they showed interesting data on MSDC-0602 use in NASH patients (Harrison, et al. J. Hepatol. 2020, 72, 613–626).

-Figure 1 legend: “b-oxidation” should be changed to beta oxidation by using a greek letter.

Author Response

We thank the Reviewer for her/his time and efforts to consider our manuscript and to offer insightful comments on our review. We re-worded the text accordingly.

All changes in the revised manuscript are in red.

Comments and answers:

-In the first paragraph of the introduction, there is discussion of pyruvate oxidation, but no mention of carboxylation. Anaplerosis is also an important mitochondrial fate of pyruvate.

A sentence has been added to mention pyruvate carboxylation (page 3 in red)

-Line 77 - the use of the word “variant” to describe MPC1-like is probably not the best choice since it is encoded by a separate gene. Also, this seems to be at odds with the discussion just above it indicating there are only two MPC genes in mammals.

We removed the word 'variant' and rephrased the sentence (see page 4, first paragraph in red)

-There should be some mention that deletion of one isoform of MPC results in the degradation of the other, i.e. knockout of one isoform essentially results in a double knockout.

This point has now been mentioned page 4, last sentence of first paragraph in red.

-The SWEET and semiSWEET schematic should be replaced with a more informative MPC topographical schematic.

We completed Figure 2 with a model for the topology of the MPC (see figure 2B, page 5)

-In the discussion of the topology, some of the more recent literature should be discussed such as Gomkale et al 2020; and Medrano-Soto 2020. It seems as though there is a growing consensus.

We have now cited these 2 papers (page 4 and 5)

-The statement that “MPC in liver is required for gluconeogenesis” (line 322) seems to be too strong. While the MPC is important for gluconeogenesis, both of the references (Ref 66, 67) show that mice are still able to maintain a degree of gluconeogenic capacity from other mechanisms.

We fully agree with the reviewer and removed this sentence.

-Recent results from the EMMINENCE trial should be included since they showed interesting data on MSDC-0602 use in NASH patients (Harrison, et al. J. Hepatol. 2020, 72, 613–626).

These results have been mentioned page 13.

-Figure 1 legend: “b-oxidation” should be changed to beta oxidation by using a greek letter.

This has been changed.